

# First characterization and validation of FORLI-HNO₃ vertical profiles retrieved from IASI/Metop

Gaétane Ronsmans[1], Bavo Langerock[2], Catherine Wespes[1], James W. Hannigan[3], Frank Hase[4], Tobias Kerzenmacher[4], Emmanuel Mahieu[5], Matthias Schneider[4], Dan Smale[6], Daniel Hurtmans[1], Martine De Mazière[2], Cathy Clerbaux[1,7], Pierre-François Coheur[1]

[1]Université Libre de Bruxelles (ULB), Faculté des Sciences, Chimie Quantique et Photophysique, Brussels, Belgium
[2]Belgisch Instituut voor Ruimte-Aëronomie–Institut d'Aéronomie Spatiale de Belgique (IASB-BIRA), Brussels, Belgium
[3]Atmospheric Chemistry Division, National Center for Atmospheric Research, Boulder, CO, USA
[4]Karlsruhe Institute of Technology (KIT), Institute for Meteorology and Climate Research (IMK-ASF), Karlsruhe, Germany
[5]Institute of Astrophysics and Geophysics, University of Liège, Liège, Belgium
[6]National Institute of Water and Atmospheric Research Ltd (NIWA), Lauder, New Zealand
[7]LATMOS/IPSL, UPMC Univ. Paris 06 Sorbonne Universités, UVSQ, CNRS, Paris, France

*Correspondence to*: Gaétane Ronsmans (gronsman@ulb.ac.be)

**Abstract.** Knowing the spatial and seasonal distributions of nitric acid (HNO₃) around the globe is of great interest to apprehend the processes regulating stratospheric ozone, especially in the polar regions. Thanks to its unprecedented spatial and temporal sampling, the nadir-viewing Infrared Atmospheric Sounding Interferometer (IASI) allows sounding the atmosphere twice a day globally, with good spectral resolution and low noise. With the Fast Optimal Retrievals on Layers for IASI (FORLI) algorithm, we are retrieving, in near-real time, columns as well as vertical profiles of several atmospheric species, amongst which is HNO₃. We present in this paper the first characterization of the FORLI-HNO₃ profile products, in terms of vertical sensitivity and error budgets. We show that the sensitivity of IASI to HNO₃ is highest in the lower stratosphere (10-20 km), where the largest amounts of HNO₃ are found, but that the vertical sensitivity of IASI only allows one level of information on the profile (DOFS~1). The sensitivity near the surface is negligible in most cases, and for this reason, a partial column (5-35 km) is used for the analyses. Both vertical profiles and partial columns are compared to FTIR ground-based measurements from the Network for the Detection of Atmospheric Composition Change (NDACC) to characterize the accuracy and precision of the FORLI-HNO₃ product. The profile validation is conducted through the smoothing of the raw FTIR profiles by the IASI averaging kernels and gives good results, with a slight overestimation of IASI measurements in the Upper Troposphere-Lower Stratosphere (UTLS) at the 6 chosen stations (Thule, Kiruna, Jungfraujoch, Izaña, Lauder and Arrival Heights). The validation of the partial columns (5-35 km) is also conclusive with a mean correlation of 0.93 between IASI and the FTIR measurements. An initial survey of the HNO₃ spatial and seasonal variabilities obtained from IASI measurements for a one year (2011) data set shows that the expected latitudinal gradient of concentrations from low to high latitudes and the large seasonal variability in polar regions (cycle amplitude around 30% of the seasonal signal, peak-to-peak) are well represented with IASI data.



## 1 Introduction

Nitric acid is the main form of oxidized nitrogen, in both the stratosphere and the troposphere, and constitutes the principal chemical sink/reservoir for $NO_x$ ($\equiv NO+NO_2$) (Austin et al., 1986; Crutzen, 1979; Wespes et al., 2009). Being directly coupled to $NO_x$, $HNO_3$ impacts the ozone ($O_3$) budgets in the two layers (e.g. Neuman et al., 2001).

In the troposphere, the main sources of $NO_x$ are fossil fuel combustion and biomass burning (~70%). Natural sources exist, such as lightning and microbial activity in soils, but their contribution to the total tropospheric $NO_x$ is smaller than the anthropogenic one, especially in industrialized areas (Cooper et al., 2014; Kasibhatla et al., 1993; Logan, 1983; Wespes et al., 2007). The distribution of the $NO_X$ sources influences directly that of $HNO_3$, which in the troposphere has a residence time of a few days to several weeks, depending on the latitudes (Logan et al., 1981; Wespes et al., 2007).

In the stratosphere, the main source of $NO_x$ is nitrous oxide ($N_2O$) which is emitted at the surface by a variety of sources, including agricultural activities (Chipperfield, 2009; McElroy et al., 1976), and is then transported to the stratosphere, where it photodissociates or reacts with $O(^1D)$ to form two NO molecules (Fischer et al., 1997; Muller, 2011; Portmann et al., 2012). The formed $NO_x$ catalyse stratospheric ozone destruction through several cycles (Mohanakumar, 2008; Solomon, 1999). Apart from being an important reservoir species for $NO_x$, $HNO_3$ is a key species for the formation of polar stratospheric clouds (PSCs, type I) during the polar winter (Höpfner et al., 2006; Lambert et al., 2012; Tabazadeh et al., 2000). PSCs, which develop at very low temperatures (195 K) and which are composed mainly of $HNO_3$, sulphuric acid and ice (Drdla & Müller, 2010; Lowe & MacKenzie, 2008), allow heterogeneous reactions and lead to the activation of chlorinated compounds in the gas phase, which in turn induce the subsequent massive destruction of ozone in the low to middle stratosphere of the polar regions in mid-spring (von Clarmann, 2013; Wegner et al., 2012). The process is amplified by the denitrification accompanying the sedimentation of the $HNO_3$-rich particles at the end of winter, which prevents the reformation of chlorine nitrate, one of the stable chlorine reservoirs (Gobbi et al., 1991; Solomon, 1999). In a denitrified stratosphere, most often observable in the Antarctic due to lower temperatures, very low $HNO_3$ concentrations are observed inside the polar vortex and a high concentration collar remains at the edge of the vortex (Santee et al., 1999; 2004; 2005; Staehelin et al., 2001; Wespes et al., 2009).

Note that the sedimentation is not the main sink for stratospheric $HNO_3$ at a broader scale; the principal degradation pathways are oxidation with the hydroxyl radical and photodissociation (Austin et al., 1986).

$HNO_3$ has been measured by a variety of instruments since its first observation from infrared solar absorption spectra in 1968 (Murcray et al., 1968). Ground-based instruments (Fiorucci et al., 2013; Rinsland et al., 1991; Wood et al., 2004), sounding instruments on board balloons or aircrafts (Jucks et al., 1999; Neuman et al., 2001) or embarked on satellites (Austin et al., 1986; Orsolini et al., 2009; Wespes et al., 2007) or aboard the space shuttle (Rinsland et al., 1996) have all contributed to the characterization of the $HNO_3$ distributions throughout the low atmosphere. One of the most complete data sets has been acquired by the Microwave Limb Sounder (MLS) first on the Upper Atmosphere Research Satellite (UARS) from 1991 to 1998, then on the AURA satellite from 2000. It has allowed detailed analyses of seasonal and interannual variations (Santee



et al. 1999; 2004) but at a coarse horizontal resolution due to the viewing mode. The vertical resolution of MLS ranges between 3 and 5 km, and the instruments probes the entire altitude range from the ground to 90 km. However, the $HNO_3$

measurements are considered reliable only in a narrow altitude range between 11 and 30 km, where the precision on the retrieved volume mixing ratio is 0.6–0.7 ppbv (Santee et al., 2007). $HNO_3$ distributions have also been obtained by the MIPAS instrument on ENVISAT, in the range 14-43 km with a sampling of 3-4 km and a reported accuracy of 0.2-0.6 ppbv (Piccolo & Dudhia, 2007; Vigouroux et al., 2007), and by the ACE-FTS on-board SCISAT with even better accuracy (3%) between 10 and 37 km (Wang et al., 2007). Measurements from the ODIN instrument made at high vertical resolution (1.5-2

60    km) but with a precision of only 1.0 ppbv over the altitude range 18-45 km (Urban et al. 2009; Wang et al., 2007) have been somewhat less used so far.

The Infrared Atmospheric Sounding Interferometer (IASI) on the MetOp satellite series operates in a different geometry (nadir) than all aforementioned instruments, and allows monitoring $HNO_3$ using spectral information from its $\nu_5$ and $2\nu_9$ vibrational bands. The IASI measurements have a limited vertical resolution because of the integrated view of the

atmospheric column but they are made at exceptional spatial and temporal sampling. Specifically, as detailed in section 2, IASI provides global measurements with a particularly good spatial and temporal sampling of the polar regions at all seasons. Other key features of IASI for $HNO_3$ are the fact that the instrument provides simultaneous measurements of $O_3$ and other trace gases, allowing studying the coupled $HNO_3$-$O_3$ cycles, and that it will operate on a long-term (2007-2022), allowing the identification and monitoring of trends. The potential of using the $HNO_3$ measurements of IASI was first shown

by Wespes et al. (2009) using a two-year data set. Although important conclusions and perspectives for the capability of IASI to sound $HNO_3$ were drawn from this study, it suffers from the fast operational retrieval method restricted to a total column calculation for the purpose of saving time, and from a missing quantitative validation due to the absence of archived data from ground-based FTIR measurements. Moreover, the total column retrieval combined with its maximum sensitivity in the stratosphere was shown to largely mask the potential of IASI to sound $HNO_3$ in lower layers. After this study, the Fast

Optimal Retrieval on Layers for IASI (FORLI) software was adapted to allow retrieval of $HNO_3$ vertical profiles from IASI (Hurtmans et al., 2012). A full 8-year data set of global profiles is now available (2008-2015) but has not yet been used for extensive analysis (a subset was used in Cooper et al. (2014) to constrain lightning $NO_x$ emissions in models). The product characterization and validation are also still lacking. This paper uses a full year (2011) of $HNO_3$ profiles retrieved from IASI on Metop-A to

• Fully characterize the $HNO_3$ retrieved concentrations in terms of vertical sensitivity and errors (section 3) on the profiles and partial columns,

• Validate the profiles and columns using correlative data from the NDACC FTIR network, which are also detailed here (section 4 and 5),

• Provide an overview of how the product can be used to analyse spatial and temporal variability (section 6).



## 2 IASI measurements

### 2.1 IASI instrument

The first IASI instrument (IASI-A) was launched in 2006 on the Metop-A platform in a polar orbit (Clerbaux et al., 2009; Hilton et al., 2012). It is still operating nominally at the time of writing, in parallel with IASI-B that was launched on Metop-B in 2012. IASI is a nadir-viewing infrared Fourier transform spectrometer measuring the radiation emitted by the Earth's surface and the atmosphere in the 645-2760 cm$^{-1}$ spectral range (August et al., 2012; C. Clerbaux et al., 2009). The spectral resolution is 0.5 cm$^{-1}$ after apodization over the entire spectral range (Cayla, 2001; Hilton et al., 2012). The apodized radiometric noise is low, around 0.2 K in the atmospheric window of interest to this work, which includes the $\nu_5 + 2\nu_9$ band of $HNO_3$ (860-900 cm$^{-1}$), mostly suitable for the retrievals, as described in Wespes et al. (2007, 2009). IASI collects 120 views every 8s along the 2200 km swath across to the satellite track and provides this way global coverage twice a day (Clerbaux et al., 2009), with one overpass in the morning and one in the evening, at 9:30 equator crossing time. The spatial resolution varies from 113 km$^2$ at nadir to 400 km$^2$ at the end of the swath.

Note that IASI measures in addition to $HNO_3$ a series of greenhouse (carbon dioxide ($CO_2$, Crevoisier et al., 2009), methane ($CH_4$, Crevoisier et al., 2013)) and reactive trace gases (carbon monoxide (CO, George et al., 2009), $O_3$ (Boynard et al., 2016; Wespes et al., 2016), ammonia ($NH_3$, Van Damme et al., 2014)), which altogether provide an extensive monitoring of the atmospheric system.

### 2.2 Retrieval method and settings

The IASI data are processed every day in near-real time at ULB, by the FORLI algorithm which relies on a fast radiative transfer and on a retrieval methodology based on the Optimal Estimation Method (OEM, Rodgers, 2000) to solve the inverse problem in the retrieval (Hurtmans et al., 2012). FORLI provides twice-daily vertical distributions of three species, namely $O_3$, CO and $HNO_3$. The FORLI methods have already been largely described (Hurtmans et al., 2012) so only a brief reminder will be presented here, focusing on the retrieval parameters for $HNO_3$.

The forward model can be written in a generic way as

$$\mathbf{Y}=\mathbf{F(x,b)}+\mathbf{\eta} \tag{1}$$

where $\mathbf{Y}$ is the measurement vector (the IASI calibrated and apodized radiances in our case), $\mathbf{x}$ is the retrieved state vector, $\mathbf{b}$ includes all parameters influencing the measurement, and $\mathbf{\eta}$ is the measurement noise. $\mathbf{F}$ is the forward function, which describes the complete physics of the measurement (Hurtmans et al., 2012; Rodgers, 2000).

The inverse problem consists in finding a state vector $\hat{\mathbf{x}}$ approximating the true state vector $\mathbf{x}$, in accordance with the measurement $\mathbf{Y}$ and with a prior knowledge of the state of the atmosphere, characterized by an a priori profile $\mathbf{x}_a$ and the corresponding variance-covariance matrix $\mathbf{S}_a$. The solution of the above equation for a linear problem is expressed as

$$\hat{\mathbf{x}} = \mathbf{x}_a + \left(\mathbf{K}^T\mathbf{S}_\epsilon^{-1}\mathbf{K}+\mathbf{S}_a^{-1}\right)^{-1}\mathbf{K}^T\mathbf{S}_\epsilon^{-1}\left(\mathbf{y}\text{-}\mathbf{K}\mathbf{x}_a\right) \tag{2}$$





where $\mathbf{K}$ is the Jacobian of the forward model $\mathbf{F}$, and $\mathbf{S}_\epsilon$ is the measurement error covariance. For a non-linear problem as ours, the solution is found iteratively. The optimal estimation provides a very appropriate framework for characterizing the retrieved profiles, in terms of vertical sensitivity (analysed with the averaging kernel functions) and errors. The way those quantities are calculated are described in Hurtmans et al. (2012) and are not repeated here. FORLI-HNO$_3$ in its latest version (v.20140922) provides profiles on 41 layers (from surface up to 40 km). The retrieval parameters, adapted from Wespes et al. (2009) and Hurtmans et al. (2012), are detailed in Table 1. The spectral range for the retrieval of HNO$_3$ profiles is 860-900 cm$^{-1}$ (see Fig.2 in Wespes et al. 2009), in which only water vapour significantly interferes. The a priori profile ($\mathbf{x}_a$) is defined as the mean of a combination of daily profiles from the LMDz-INCA chemistry-transport model (from the ground up to 15.6 km) and of all profiles obtained from ACE-FTS (from 6 to 60 km). The resulting mean profile is constant over time and does not depend on latitude or longitude, i.e. the same a priori profile is used for all observations around the globe. A variance-covariance matrix from the ensemble of profiles is then calculated ($\mathbf{S}_a$) and yields high variability in the boundary layer (170%) and in the UTLS region (80%) and lower variability in the troposphere and the stratosphere (50% and 20%, respectively). The uncorrelated noise varies around the value of $2.10^{-8}$ W/(cm$^2$cm$^{-1}$sr). The FORLI-HNO$_3$ retrieval performances in terms of root mean square (RMS) and bias values of the spectral residuals calculated after the retrievals, of retrieval total errors, and of degree of freedom for signal (DOFS calculated as the trace of the averaging kernel matrix – DOFS=trace(A)) are detailed in Hurtmans et al. (2012). Identically to Wespes et al. (2009), posteriori filtering of the data has been performed to remove some strongly biased HNO$_3$ observations. For instance, only HNO$_3$ observations with a good spectral fit (RMS of the spectral residual lower than $3.10^{-8}$ W/(cm$^2$cm$^{-1}$sr)) have been analysed. Additional quality flags rejecting biased or sloped residuals, suspect averaging kernels and maximum number of iterations exceeded are also applied. Cloud contaminated IASI scenes are also filtered out, based on cloud information from the Eumetcast operational processing (fractional cloud cover below 25%).

## 3 Characterization of FORLI-HNO$_3$ profiles and columns

Figure 1 displays for illustration typical FORLI-HNO$_3$ retrieved profiles at tropical (Izaña), mid (Jungfraujoch) and polar (Thule) latitudes in July as well as the a priori profile. It can be seen that the tropospheric concentrations are small and remain close to the a priori for all latitudes. Only Izaña stands out in this respect, with the retrieved concentrations 30% smaller than those of the a priori profile. The stratospheric concentrations are much higher, especially between 15 and 25 km altitude, and the better sensitivity of the instrument at these altitudes (discussed next) allows for significant departure from the a priori profile. For these three example profiles, maximum values range between less than 5 ppbv at Izaña and 9.5 ppbv at Thule. This latitudinal difference is in fact a persistent and well-documented feature (e.g. Santee et al., 2007, 1999; Wespes et al., 2007), which is mostly associated with reduced HNO$_3$ photodissociation in the polar regions, mainly in winter.



The averaging kernels, presented in Figure 2 (top panels) for the same three locations as in Figure 1, allow characterizing the sensitivity of IASI to the $HNO_3$ profile, within the general FORLI framework. We find that the averaging kernels show similar shapes whatever the latitude and cover the whole range of altitudes from the surface to the upper stratosphere. The limited amount of vertical information is well seen from the overlap between the individual layer kernels. Also the quasi absence of sensitivity in the low troposphere below 5 km is obvious, with absolute values of the averaging kernels close to zero. This confirms the conclusion from Wespes et al. (2009) that the IASI instrument does not carry several pieces of vertical information for $HNO_3$. From the averaging kernels, we can also conclude that the maximum sensitivity to $HNO_3$ is at around 15 km, slightly higher (18 km) at tropical latitudes in comparison to polar latitudes (13 km). This translates to a maximum sensitivity in the upper troposphere-lower stratosphere (UTLS) at equatorial latitudes, and in the low to middle stratosphere at polar latitudes, which corresponds generally to the altitude of highest concentrations.

In order to have a global vision of the results, Figure 3 shows the global distributions of the degrees of freedom for signal (DOFS, top panels) separately for January (left) and July (right) 2011. These represent the number of independent pieces of information in the measurements and give an estimation of the vertical sensitivity of the retrievals. On average, all DOFS values are close to 1, further indicating that only one level of information can be extracted from the IASI data for $HNO_3$. However, there are some latitudinal differences, with the DOFS being generally larger in the intertropical belt, with values around 1.1 or slightly more (e.g. in the deserts during the summer) due to larger temperatures inducing a better signal to noise ratio, in comparison to the mid and polar latitudes, where the DOFS is mostly around 0.9. However, it should be noted that the larger values of DOFS, particularly the ones found in the deserts, might also be, at least partly, attributed to the misrepresentation of the emissivity of these surfaces (Hurtmans et al., 2012). Indeed, even though we also find high surface temperatures (see Figure 3, middle panels) in most of the intertropical region of the globe, it does not necessarily induce such a large DOFS value. The lowermost panels in Figure 3 depict the altitude of maximum sensitivity of the IASI in January (left) and July (right). We show that the altitude of maximum sensitivity is invariant at equatorial and tropical latitudes, whereas it varies with the season at mid- and polar latitudes. As expected, the variations of the altitude of maximum sensitivity with the seasons follow the maximum $HNO_3$ concentrations (see below).

The optimal estimation method also enables characterizing the retrieved profiles in terms of error profiles. The total error on the retrieved profile can be divided into three components (Hurtmans et al., 2012; Rodgers, 2000):

- the smoothing error, due to the consideration that the retrieved profile is an estimation of a state smoothed by the averaging kernel, rather than an estimate of the true state,
- the error on the model parameters, due to fixed parameters in the direct model, e.g. surface emissivity, temperature profile. This error is not taken into account in the routine processing of the error matrix in FORLI,
- the error due to the radiometric noise, i.e. the measurement error.

The contributions of the different error sources are presented in Figure 4. This plots shows that the smoothing error is by far the main source of retrieval error over the entire altitude range (Wespes et al., 2007). We find, in agreement with the low



values of the averaging kernels, that the total retrieval error in the low troposphere reproduces the a priori covariance, meaning that no information on $HNO_3$ profile is obtained from the IASI measurement at these altitudes. The total error decreases with increasing altitude, reaching minimum values of about 20% around 20 km, where the sensitivity is highest.

Comparing with the a priori covariance, the precision is in fact increased mainly between 5 and 25 km, and especially around 14 km where the reduction of error reaches almost 50%. Above 30 km, the gain of information is again minimal. The measurement error is a minor source of error; it reaches a maximum of 30% in the boundary layer and quickly becomes negligible above 10 km. The error on the profile translates to a total retrieval error in the range of 5 to 50% depending on latitude, with a mean value of 10%. When calculated for various partial columns (see box in Figure 4, left), it appears clearly

that the tropospheric column carries the largest error (62%), whereas the error is lower for the total (10%) or the stratospheric (8%) column. Following the analysis of the error profiles in Figure 4 (left), a column ranging from 5 to 35 km can be considered as the one carrying most of the retrieved information; it is characterized by a total retrieval error of 3% on a global average and the DOFS is the same as for the total column (ranging from 0.9 to 1.2).

Spatially, we find that the total error is larger at tropical latitudes, with values around 10-15% (Figure 4, right), mainly due to

195 the higher concentrations of water vapor, which has absorption lines that interfere under large humidity with the ones of $HNO_3$ in the spectral region used for the inversion (Hurtmans et al., 2012; Wespes et al., 2009). The mid and polar latitudes are characterized by much lower total retrieval errors, with maximum values of about 3% for the 5 to 35 km column. This corresponds to a reduction by a factor 30 as compared to the prior uncertainty defined by $\mathbf{S}_a$ (90%). The very large errors found in Antarctica are most probably due to a misrepresentation of the surface emissivity above cold regions and to a very

poor sensitivity above such cold regions (Hurtmans et al., 2012).

## 4 Validation methodology

The IASI derived $HNO_3$ profiles/columns are compared here with reference profiles/columns retrieved from measurements made by ground-based infrared Fourier-transform spectrometers (FTIR) for the sake of validation. Note that the FTIR $HNO_3$ profiles have been used to validate other satellite datasets before (Vigouroux et al., 2007; Wang et al., 2007; Wolff et al.,

2008; Wood et al., 2002).

### 4.1 FTIR stations and instrument

For our study, six stations routinely operating FTIR instruments, which are all part of the Network for the Detection of Atmospheric Composition Change (NDACC, http://www.ndacc.org) and which span a large range of latitudes, were selected for the validation: Thule (76.5°N, 69°W), Kiruna (67.8°N, 20.4°E), Jungfraujoch (46.6°N, 8.0°E), Izaña (28.3°N; 16.5°E),

Lauder (45.0°S, 169.7°E) and Arrival Heights (77.8°S, 166.7°E). The location of each station is displayed in Figure 3 (red dots, upper left panel) and details on the operating instrument and retrieval algorithms are given in Table 2. The FTIRs



provide data year round for all six stations, except at high latitudes (Thule and Arrival Heights in our case) where the lack for light prevents solar measurements during winter.

All instruments are Bruker spectrometers (120M for Thule and Arrival Heights, 120HR for Jungfraujoch and Lauder, and

125HR for Kiruna and Izaña). The algorithms used for the retrieval of $HNO_3$ profiles are PROFFIT (Hase et al., 2004) for Kiruna and Izaña, SFIT2 (Pougatchev al. 1995; Rinsland et al., 1998) for Thule and Jungfraujoch, and SFIT4 (Pougatchev et al., 1995) for Lauder and Arrival Heights. An updated Lauder and Arrival Heights $HNO_3$ data set was used in this study. The Lauder and Arrival Heights retrievals implemented SFIT4 closely adhering to the prescribed NDACC IRWG $HNO_3$ retrieval strategy (see http://www.acom.ucar.edu/irwg/IRWG_Uniform_RP_Summary-3.pdf), with the following particularities: at

both sites the spectral resolution is 0.0035 cm$^{-1}$ and pressure and temperature profiles (ZPT) were obtained from NCEP. Using LINEFIT (Hase et al., 1999) to diagnose the instrument lineshape (ILS), an ideal ILS is assumed at Lauder whilst a parametrised linear ILS from 1.0 at ZPD to 0.95 (max OPD) is used in Arrival Heights retrievals. A single beam channelling fit is also implemented in the Arrival Heights retrievals. The signal-to-noise ratio (SNR) is calculated per spectrum and yields typical values of 180 and 195 for Lauder and Arrival Heights, respectively.

The three algorithms (PROFFIT, SFIT2 and SFIT4) use the optimal estimation method (OEM) developed by Rodgers (2000) which facilitates the comparison with IASI. All stations use microwindows in the region 866-875.2 cm$^{-1}$ for the retrieval of $HNO_3$ profiles and the DOFS range between 1.9±0.5 (Jungfraujoch) and 3.1±0.4 (Thule). The a priori profiles are described individually at each station, regardless of seasonality. It should be noted that, due to the lower degrees of freedom in the Lauder and Arrival Heights retrievals, there were numerical artefacts in the smoothing operation between the IASI and the

FTIR ground-based measurements (see section 4.2). To reduce these artefacts, a smoothed IASI a priori profile was used as the a priori in the ground-based retrievals in this study. The associated ground-based retrieval a priori covariance matrices ($S_a$) were also constructed from a smoothed IASI covariance data set. Spectroscopic line parameters were all taken from HITRAN 2008 database (Rothman et al., 2009). Note that IASI retrievals use spectroscopic line parameters taken from HITRAN 2004 database (Rothman et al., 2005). The update consists mainly in an improvement of the line positions and

intensities (Flaud et al., 2006; Gomez et al., 2009; Rothman et al., 2009) and the differences it might induce should be kept in mind when analysing the comparison between the two instruments. Additional differences between the two instruments might also come from the uncertainty of the FTIR measurements themselves, which is dominated by the temperature and the spectroscopy information in the retrieval.

## 4.2 Co-location criteria and comparison method

The validation is performed for the year 2011 of IASI data. For the spatial co-location, the line of sight of the FTIR measurement is calculated, as well as the point along that line where the sensitivity is highest (i.e. where the total averaging kernel (see Figure 2, black dashed line) reaches a maximum). A large 'box' in which IASI measurements might be considered was defined for each station (approximately 10° of latitude and 15° of longitude around each station), but it is the



location of the maximum sensitivity along the line of sight that determines the reference for the co-location with IASI

measurements. The co-location criteria chosen is such that the IASI measurements should be within 0.5° in latitude and 1° in longitude from the FTIR reference point.

The co-location in time has been chosen after several tests (not shown) as ≤ 12h. If more than one IASI measurement was satisfying these criteria, then all IASI retrieved profiles inside that spatial and temporal window were averaged. Each pair (FTIR, co-located IASI) undergoes the validation steps described below and in Rodgers & Connor (2003).

The raw profiles retrieved from IASI and FTIR cannot be compared adequately because of the difference in vertical sensitivity (Rodgers & Connor, 2003). To illustrate this, Figure 2 depicts both IASI and FTIR typical averaging kernels (top and bottom panels, respectively) as well as the IASI and FTIR so-called "sensitivities" (red curves). The sensitivity at altitude $i$ is calculated as the sum of the elements of the corresponding averaging kernel, $\sum_j A_{ij}$ (with $A$ the averaging kernels matrix) and represents the fraction of the retrieval that comes from the measurement rather than from the a priori

profile (Vigouroux et al., 2007). As opposed to this sensitivity, the total averaging kernel (dashed black lines) is calculated as $\sum_i A_{ij}$ and represents the contribution of each level to the sensitivity at a given altitude $i$.

We find that the FTIR instrument provides better vertical resolution, with each averaging kernel maximum close to its corresponding layer, and a total DOFS within the known values (see Table 2) for these examples. The averaging kernel associated with the total column (Figure 2, black dashed lines) show a maximum around 200 hPa for both instruments at all

latitudes. Looking at the sensitivity of the two instruments (red lines), two maxima reaching 1 appear in the Thule FTIR example; one around 200 hPa (same as IASI) and one around 20 hPa, indicating that the measurement is specifically sensitive to variations in $HNO_3$ concentrations in those two regions of the atmosphere. On the other hand, the IASI maximum sensitivity largely exceeds the value of 1 around 200 hPa at all latitudes, indicating that the instrument might be over-sensitive to $HNO_3$ concentrations in that region of the atmosphere (Vigouroux et al., 2007), and that it might over-

compensate the lack of sensitivity in other regions of the atmosphere, yielding large $HNO_3$ concentrations.

In order to compare an FTIR-IASI profiles pair and to reduce the smoothing uncertainties on the comparison, we use Rodgers and Connor (2003). First we insert the IASI a priori in the FTIR profile using

$$\mathbf{x}_{fi} = \mathbf{x}_f + (\mathbf{A}_f - I)(\mathbf{x}_{af} - \mathbf{x}_{ai}) \tag{3}$$

where $\mathbf{x}_{fi}$ is the FTIR profile aligned with the IASI a priori $\mathbf{x}_{ai}$, $\mathbf{x}_f$ is the raw FTIR profile, $\mathbf{x}_{af}$ the a priori FTIR profile, and

$\mathbf{A}_f$ is the FTIR averaging kernel matrix.

Next, this aligned FTIR profile is interpolated to the IASI altitude grid and the standard smoothing equation is applied, using the IASI averaging kernel and a priori:

$$\mathbf{x}_s = \mathbf{x}_{ai} + \mathbf{A}_I \left( \mathbf{x}_{fr} - \mathbf{x}_{ai} \right) \tag{4}$$

where $\mathbf{x}_s$ is the smoothed version of $\mathbf{x}_f$, $\mathbf{x}_{fr}$ is the regridded FTIR profile and $\mathbf{A}_I$ is the IASI averaging kernels matrix. A

final step is to average, for a given FTIR profile, all corresponding IASI and smoothed FTIR profiles. This way, FTIR measurements with a high number of co-locations do not influence the statistics.



The comparison is carried out next for the profiles (section 5.1) or partial columns (5-35 km) (section 5.2). For the discussion, we rely on the relative difference between the quantities retrieved from IASI and the corresponding ones from the smoothed FTIR data following:

$$x(\%) = \frac{IASI - FTIR}{FTIR} \times 100 \qquad (5)$$

with $x$ being the relative difference. Standard deviations are also calculated as:

$$\sigma_x = \sqrt{\frac{1}{N-1} \sum_{i-1}^{N} (x_i - \bar{x})^2} \qquad (6)$$

with $N$ the number of observations, $x_i$ the difference between IASI and FTIR values for the $i^{th}$ observation and $\bar{x}$ the mean of the differences for all measurements, and allow the characterization of the variability of the data set.

## 5 Validation results

### 5.1 Vertical profiles

The comparison of the $HNO_3$ mean vertical profiles retrieved from IASI and the FTIRs for each station is presented in Figure 5, with the six panels on the left showing the comparison between IASI profile (red) and the raw (black) and smoothed (green) FTIR profiles, as well as IASI and FTIR a priori profiles (dashed red and dashed black lines, respectively). The right panel shows the mean relative differences (%) between IASI and the smoothed FTIR profile for each station. For all stations, the raw FTIR profile is largely different from the IASI profile, and this difference remains after regridding on the IASI retrieved levels (not shown). The smoothing of the FTIR profile with IASI averaging kernels brings the two profiles much closer together, with differences generally below 50% at all altitudes and maximum differences in the upper troposphere-lower stratosphere between 300 and 50 hPa. At the higher latitudes, the differences in the profiles are always below 20%. The statistics of the comparison between profiles is summarized in Table 3. Overall, the troposphere and higher stratosphere record lower values of relative difference between IASI and the smoothed FTIR profile. However, this is in most part related to the low vertical sensitivity in these regions of the atmosphere, forcing the retrieval to rely mostly on the a priori information, i.e. the same information after the smoothing of the FTIR profiles with IASI a priori information. As indicated in the Table 3 and as can be seen in Figure 5 (right), at all stations, the maximum values for the relative differences with the smoothed profile are located in the low stratosphere (around 13 km altitude), where the sensitivity of IASI to the measurement is the largest (see Fig.2, top panels). These differences are always positive, suggesting an overestimation of the IASI concentrations in that region of the atmosphere, compared with the FTIR data, and they vary with the latitude (from 4.7% at Arrival Heights to 47.2% at Lauder). Important differences between IASI and the FTIR are also found in the boundary layer, especially in Kiruna (-11.8%), Jungfraujoch (19.7%) and Arrival Heights (43.9%) and probably do not reveal a real difference, but rather an artefact due to the regridding with potential differences in the altitude taken as ground level in both instruments. It should be noted that an overestimation of IASI measurements compared with ground-based





measurements in the upper troposphere-lower stratosphere has also been found for $O_3$ vertical profiles (Antón et al., 2011; Dufour et al., 2012; Gazeaux et al., 2013). While some hypotheses have been brought forward by Dufour et al. (2012), the exact reason for that particular feature of FORLI for both $HNO_3$ and $O_3$ retrievals is not clear.

**5.2 Partial columns**

$HNO_3$ partial columns from 5 to 35 km have then been compared in a similar way as the profiles. We recall here that the choice of this partial column is made in order to consider only the range of altitudes where both instruments are sensitive (see discussion in section 3 and comparison of averaging kernels in Figure 2), getting rid of the low troposphere and the high stratosphere. The results for the 2011 time series are displayed in Figure 6. For each station, the top panel shows the
comparison between raw (black) and smoothed (green) FTIR and IASI partial columns (red). Each grey point represents a daily mean of all IASI observations recorded in the area around the FTIR station (the predefined 'box' around each station, see section 4.2.), with a 3σ confidence interval around each grey point. The entire time evolution of the columns from IASI is displayed in grey. It shows the exceptional temporal sampling and especially the usefulness of IASI measurements in the winter months at high latitude. The IASI a priori is also represented (grey dashed line) for each station. Note that the slight
temporal variability of the IASI a priori is due to its representation in column units, making it dependent on the air column at the time of measurement. The possibility to investigate chemical and physical processes from these time series is briefly explored in section 6. The bottom panels in Figure 6 show the relative differences between the different data sets. The statistics of the comparison are provided in Table 4. For all stations and all measurements, we find that the FTIR column values are within the IASI total retrieval error range (see red error bars), with mean values of $HNO_3$ partial columns very
similar between the two instruments (see Table 4, columns 5 and 6). Considering the smoothed columns, IASI is always positively biased, with bias values between 4.0% in Thule and 21.7% for Lauder, and with an overall bias (all stations together) of 11.5% (Table 4). The smoothing of the FTIR data is particularly efficient for the comparison with IASI observations at Izaña, where the bias decreases from 21.3% to 9.2% from the unsmoothed to the smoothed FTIR. Being the station with the largest difference between the FTIR and the IASI a priori profiles (see Figure 5), Izaña is a good example of
the influence of the a priori profiles in the retrievals. Indeed, with initially very different a priori profiles and limited vertical sensitivities, yielding large relative differences between the two data sets (see Figure 6, black dots), the smoothing of the FTIR data set largely decreases the mean bias (green dots).

Looking at all stations, the standard deviation of the differences is larger than the bias at Thule, Kiruna, Izaña and Arrival Heights, suggesting that the bias is non-significant compared to the variability (Kerzenmacher et al., 2012). The reason of the
different behaviour for the Jungfraujoch and Lauder mid-latitude stations is unclear at this point.

Figure 7 summarizes the results from the partial column validation, by showing (top) the correlation between IASI and FTIR data, for all stations and (bottom) the relative differences in the columns. The stations are identified by a different color. We find that the overall correlation coefficient reaches 0.93 (the correlation coefficients for each station are specified in Table 4,





last column) showing that IASI captures very well the dynamic range of variability for the column, which varies between
340 $0.9.10^{16}$ and $4.5.10^{16}$ molec.cm$^{-2}$. This high correlation coefficient must however be considered cautiously, because of the influence of the remaining a priori information, albeit moderately since the least sensitive regions have been removed by considering the 5-35 km partial column. The relative differences clearly illustrate the positive bias of IASI described above, but also that the bias does not show any particular temporal trend.

## 6 Spatial and temporal variability

### 6.1 Variability at NDACC stations

In Thule, IASI HNO$_3$ columns (red dots on Figure 6) range between $1.6 \ 10^{16}$ in summer and $4.5 \ 10^{16}$ molec.cm$^{-2}$ in spring. The amplitude of the seasonal cycle thus amounts to $3.0 \ 10^{16}$ molec.cm$^{-2}$. The annual cycle is consistent with the known variability of stratospheric polar latitudes (Wespes et al., 2009). The comparison of the seasonality with the FTIR is not possible at Thule since the FTIR instrument, which operates with solar light, does not provide data before March and after
350 October.

In Kiruna, IASI partial columns range between around $1.5 \ 10^{16}$ molec.cm$^{-2}$ and $3.5 \ 10^{16}$ molec.cm$^{-2}$. The winter months (February to April, approximately) are characterized by higher columns whereas the summer columns are lower (July and August), giving a seasonal amplitude of $2 \ 10^{16}$ molec.cm$^{-2}$. Due to the relatively high latitude of the station, January and December data are also missing in the FTIR record. With the data available, we find good agreement on the seasonality
between IASI and the FTIR, however with a time-dependent bias and especially a significant positive bias of IASI (~30%) for several days around mid-February and a low bias around -20% throughout March.

At Jungfraujoch and Izaña, IASI partial columns are lower than at higher latitudes, ranging between around $1.3 \ 10^{16}$ and $3.0 \ 10^{16}$ molec.cm$^{-2}$ for Jungfraujoch, and $0.9 \ 10^{16}$ and $1.7 \ 10^{16}$ molec.cm$^{-2}$ for Izaña. The amplitude of the seasonal cycle is thus much weaker than at higher latitudes ($1.7 \ 10^{16}$ molec.cm$^{-2}$ at Jungfraujoch and $0.9 \ 10^{16}$ molec.cm$^{-2}$ at Izaña), with only
360 slightly higher concentrations observed in January and February at Jungfraujoch.

In the Southern hemisphere, at Lauder, IASI partial columns range between $1.0 \ 10^{16}$ and $2.6 \ 10^{16}$ molec.cm$^{-2}$ with slightly higher values recorded during the local winter and spring (August-September, mainly). The annual cycle amplitude reaches $1.5 \ 10^{16}$ molec.cm$^{-2}$ and reflects well the small annual cycle usually recorded for southern mid-latitudes.

As for Arrival Heights, columns range between $0.9 \ 10^{16}$ and $3.1 \ 10^{16}$ molec.cm$^{-2}$, and the day-to-day variability caused by
365 the variability of the vortex itself is well seen in the data (see for example the three data points in April). Due to its very southern location and the absence of solar light during a long period of time, the FTIR instrument does not take any measurements from April until September. Given the little amount of FTIR data thus recorded, it is quite difficult to establish any clear seasonality about HNO$_3$ columns. The complete IASI data set (grey dots in Figure 6) allows assessing the seasonal cycles at all stations. It is thus particularly helpful for polar regions (Thule and Arrival Heights) where the dynamics



during winter is important. Hence, we find lower concentrations than what could be expected at Thule from January to April (consistent with the few FTIR data available in March) and a strong decrease in $HNO_3$ concentrations in June at Arrival Heights, which are discussed below. Another important feature highlighted by the complete IASI data set is the large spatial variability recorded at high latitudes. Indeed, with each grey data point representing a daily mean of all IASI observations in a quite large box around the station, and the red dots being the closest IASI observation to the FTIR reference point for the

comparison (see section 4.2. for details), the difference between the two shows the spatial variability that can exist within a defined region. Such a feature can be observed at Arrival Heights, Thule and Kiruna, whereas other stations located at more mid-latitudes (Lauder, Izaña and Jungfraujoch) show very little to no spatial variability. The daily variability is also highlighted in Figure 6, with the grey shaded areas representing the standard deviation ($3\sigma$) of the daily mean IASI observations. This variability includes the spatial variation of $HNO_3$ inside the selected boxes around the FTIR stations and

the daily variation captured by the IASI daytime and nighttime measurements. It is particularly interesting in the mid- and high latitude stations, especially in winter (Januay-April for Thule, July-September for Arrival Heights), where it is much larger than the error values (~$1.7.10^{16}$ molecules.cm$^{-2}$ for the daily variations *vs* ~$0.5.10^{16}$ molecules.cm$^{-2}$ for the total error - red vertical error bars) and thus proves that the IASI instrument captures a real daily variability of $HNO_3$ partial columns at these latitudes. This is mainly linked to the rapid zonal transport of air masses induced by the development of the polar

vortex in these regions (Wespes et al., 2009). Regarding the tropical latitudes, the daily variability should be considered more cautiously, since it is of the same magnitude as the retrieval error. Though not in the scope of the present study, the question of day and night variability needs to be further investigated.

**6.2 Global variability**

Beyond assessing the validity of $HNO_3$ locally, IASI offers the potential of global analysis thanks to its sampling. Monthly

global distributions of the $HNO_3$ total columns in 2011, calculated from the FORLI-$HNO_3$ retrieved vertical profiles are shown in Figure 8 for the Northern (left) and Southern (right) Hemispheres. On the other hand, Figure 9 depicts mean daily $HNO_3$ concentrations time series for 9 latitudinal bands of 20° each (Northern Hemisphere on the left, Southern Hemisphere on the right). From these two figures, we show that the general spatial features with lower columns at tropical latitudes and higher columns at polar latitudes are well seen, especially during the winter.

The tropical regions show very low column values (around $1.0\ 10^{16}$ molec.cm$^{-2}$) all year round. Identically to the $HNO_3$ total column product (Wespes et al., 2009) no seasonality is observed (average amplitude of only $2.3\ 10^{15}$ molec.cm$^{-2}$ for Equator and both Tropics), nor any particular spatial pattern, as can be seen from Figures 8 and 9. The mid- and polar latitudes in both hemispheres record higher concentrations, especially during the winter, with maximum values of up to $4.5\ 10^{16}$ molec.cm$^{-2}$ (Arctic). The build-up of high $HNO_3$ concentrations starts at the beginning of the winter (October) and lasts until

April-May, where the longer days enhance photodissociation. In the southern hemisphere, the concentrations also increase at the beginning of winter (April), but decrease rapidly (within one month, see June in Figure 8 (right) and dark blue in Figure



9 (right)) with the denitrification process happening during winter. Due to denitrification, only a high concentration collar remains at the vortex edge, while the concentrations inside the polar vortex drop to as low as 1.0 $10^{16}$ molec.cm$^{-2}$. The phenomenon is very obvious in Antarctic in July and August but, albeit moderately, also in the Arctic in January (see Figure 9 (left), light blue curve). Note that in the Arctic this is not a recurring annual feature. In 2011, it was caused by the exceptional stratospheric conditions during winter, which led also to a strong Arctic ozone depletion. This denitrification was documented before in Manney et al. (2011) based on MLS limb observations.

With these fluctuations, the polar regions record not only the highest columns but also the largest cycle amplitudes, which are around 1.0 $10^{16}$ and 1.4 $10^{16}$ molec.cm$^{-2}$ for North and South poles, respectively, with low columns in the summer and high columns in the winter.

In addition to the seasonal cycles, significant daily variability represented by the colored shaded areas in Figure 9 (calculated as 1 sigma of the daily IASI measurements) can be revealed thanks to the IASI sampling, especially at high latitudes during the denitrification periods. This daily variability has already been reported by Wespes et al. (2009). The main reason behind the day-to-day variability at high latitude is the variability of the vortex itself.

## 7 Conclusions and perspectives

In this paper we have characterized, validated and analysed the first results of the FORLI-HNO$_3$ vertical profiles data set retrieved from IASI/MetOp. The profiles are retrieved on a 41 km altitude grid twice a day, globally. A 8-year record is now available on request and an implementation in the EUMETSAT IASI-Level2 Product Processing Facility (August et al., 2012) is foreseen. One year (2011) of data has been investigated here. We have shown that IASI has a maximum sensitivity to the HNO$_3$ profile in the stratosphere, around 10-20 km altitude and that the vertical sensitivity of the instrument typically allows retrieving a single piece of information on the profile (DOFS varying from 0.9 to 1.2). The altitude of maximum sensitivity corresponds to the region of the atmosphere with the highest concentrations. The averaging kernels and error profiles showed that most of the available information in the IASI measurements originates from the altitude range between 5 to 35 km altitude. In terms of the corresponding partial column (5-35 km), the total retrieval error was calculated to be around 3% at high latitudes, where water vapour does not interfere with HNO$_3$ absorption lines, increasing to 10-15% at equatorial latitudes.

The validation was conducted by comparing the IASI retrieved HNO$_3$ profiles or partial columns to those retrieved from ground-based FTIR measurements made at six different stations spread around the globe at representative latitudes, namely Thule, Kiruna, Jungfraujoch, Izaña, Lauder and Arrival Heights. We found good general agreement between IASI and the smoothed FTIR profiles, with, however, an overestimation of IASI data compared with FTIR measurements (11.5% positive bias). In most cases, the differences were not found significant compared to the variability. The correlation between the two data sets is high for all stations (0.93 for all stations together), demonstrating the capability of IASI to capture the spatial and temporal patterns of the HNO$_3$ variability. However, as could be highlighted by the case of Izaña, the influence of the a



priori profile to the validation process is quite large, since the application of a common a priori profile to both measurements
largely improves the comparison. It should also be noted that the differences observed between the two data sets can also be
attributed, at least partly, to the difference in the spectral region used by each instrument, the different line parameters
(HITRAN 2004 for IASI and HITRAN 2008 for FTIR) and codes used for the retrievals and the errors of each instrument.

IASI data allows remarkable monitoring of year-round $HNO_3$ concentrations. The global distribution acquired with IASI
showed for instance a clear latitudinal gradient, with low and relatively constant concentrations at tropical latitudes, and
much higher and very variable concentrations at mid and polar latitudes. The daily variability also highlighted by the IASI
data set can be further investigated thanks to the capability of the IASI sounder to monitor the atmosphere at both day and
night time.

The polar processes, including the strong denitrification in Antarctic, are well monitored with IASI, both spatially and
temporally. Overall, the results presented here are extremely encouraging with regard to the use of the $HNO_3$ dataset from
IASI to investigate stratospheric processes on local to global scales, with particular interest for the polar regions. The long
time series that are available from IASI, which will span more than 15 years and which will be extended with the IASI-NG
instrument on EPS-SG (Clerbaux & Crevoisier, 2013; Crevoisier et al., 2014), will be important to monitor longer-term
changes in stratospheric composition and its link to climate.

*Acknowledgements*. IASI has been developed and built under the responsibility of the "Centre National d'Etudes Spatiales"
(CNES, France). It is flown on board the MetOp satellites as part of the EUMETSAT Polar System. The IASI L1 data are
received through the EUMETCast near-real-time data distribution service. The research was funded by the F.R.S.-FNRS, the
Belgian State Federal Office for Scientific, Technical and Cultural Affairs (Prodex arrangement 4000111403 IASI.FLOW)
and EUMETSAT through the Satellite Application Facility on Ozone and atmospheric Chemistry Monitoring (O3MSAF). G.
Ronsmans is grateful to the "Fonds pour la Formation à la Recherche dans l'Industrie et dans l'Agriculture" of Belgium for a
PhD grant (Boursier FRIA). C. Clerbaux is grateful to CNES for financial support. We thank U. Raffalski and P. Voelger for
technical support at IRF Kiruna. We would like to thank Antarctica New Zealand and the Scott Base staff for providing
logisitical support for the NDACC-FTIR measurement program at Arrival Heights. Measurements at Lauder and Arrival
Heights are core-funded by NIWA through New Zealand's Ministry of Business, Innovation and Employment. The
University of Liège contribution has been primarily supported by the F.R.S.-FNRS, the Fédération Wallonie-Bruxelles and
the GAW-CH program of MeteoSwiss. We further acknowledge the International Foundation High Altitude Research
Stations Jungfraujoch and Gomergrat (HFSJG, Bern). E. Mahieu is Research Associate with F.R.S.-FNRS. The ground-
based FTIR data used here are available from NDACC database (ftp://ftp.cpc.ncep.noaa.gov/ndacc/station/), except Lauder
and Arrival Heights, which can be requested upon demand.



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





Table 1. Retrieval settings used for the retrieval of HNO$_3$ concentration profiles, using the FORLI-HNO$_3$ software, and updated from Wespes et al. 2009. From first to last row: spectral range, a priori profile ($x_a$), a priori covariance matrix ($S_a$), uncorrelated noise ($\sigma_\epsilon$), and state vector elements.

| | |
|---|---|
| Retrieval spectral range | 860-900cm$^{-1}$ |
| $x_a$ | LMDz-INCA (ground-15.6km), ACE-FTS (6-60km) |
| $S_a$ | 0-5km: 170% ; 5-10km: 50% ; 10-20km: 80% ; 20-41km: 20% |
| $\sigma_\epsilon$ | 2.10$^{-8}$ W/(cm$^2$cm$^{-1}$sr) |
| State vector | Surface temperature, HNO$_3$ profile, Water vapour column |

Table 2. NDACC stations selected for the HNO$_3$ validation, and their location, coordinates and altitude (in meters above sea level). The instrument and retrieval code for the retrieval of the data are specified for each station, as well as the microwindows and DOFS for total columns. References for further details are listed in the first column

| Station (References) | Location | Coordinates | Altitude (m a.s.l.) | Instrument | Retrieval code | Microwindows (cm$^{-1}$) | DOFS |
|---|---|---|---|---|---|---|---|
| **Thule** (Hannigan et al., 2009) | Greenland | 76.5° N, 69° W | 225 | Bruker 120M | SFIT2 | 867.5-870.0 | 3.1 ± 0.4 |
| **Kiruna** (Blumenstock et al., 2006) | Sweden | 67.8° N, 20.4° E | 419 | Bruker 125HR | PROFFIT9 | 867.0-869.6 872.8-875.2 | 3.0 ± 0.4 |
| **Jungfraujoch** (Mahieu et al., 1997; Zander et al., 2008) | Switzerland | 46.6° N, 8.0° E | 3580 | Bruker 120HR | SFIT2 | 868.5-870.0 872.25-874.0 | 1.9 ± 0.5 |
| **Izaña** (García et al., 2012; Schneider et al., 2005) | Canary Islands | 28.3° N, 16.5° W | 2367 | Bruker 125HR | PROFFIT9 | 867.0-869.6 872.8-875.2 | 2.3 ± 0.3 |
| **Lauder** | New Zealand | 45.0° S, 169.7° E | 370 | Bruker 120HR | SFIT4 | 867.05-870.0 872.25-874.0 | 2.1 ± 0.3 |
| **Arrival Heights** | Ross Island, Antarctica | 77.8° S, 166.7° E | 200 | Bruker 120M | SFIT4 | 867.05-870.0 872.25-874.0 | 1.9 ± 0.4 |



**Table 3.** For each station: minimum and maximum values of relative differences (%) and correlation coefficients between HNO$_3$ IASI profiles and FTIR smoothed profiles. The value between brackets is the altitude (km) of minimum (maximum) relative difference between profiles.

|  | Minimum (%) [altitude (km)] | Maximum (%) [altitude (km)] |
|---|---|---|
| Thule | 0.4 [22] | 12.5 [13] |
| Kiruna | -0.1 [24] | 18.0 [13] |
| Jungfraujoch | 0.1 [37] | 25.8[12] |
| Izaña | 0.21 [2] | 45.0 [13] |
| Lauder | 1.2 [39] | 47.2 [12] |
| Arrival Heights | 0.4 [4] | 4.7 [13] |

**Table 4.** For each station: mean of relative differences (bias) in %, calculated following Eq.5 for smoothed FTIR partial columns, standard deviation of differences between IASI and smoothed FTIR, mean values for both data sets, number of comparison pairs and correlation coefficient between IASI and smoothed FTIR.

| Stations | Bias (%) Smoothed FTIR | Standard deviation (%) | Mean IASI (molec.cm$^{-2}$) | Mean FTIR (molec.cm$^{-2}$) | # pairs | R |
|---|---|---|---|---|---|---|
| **Thule** | 4.0 | 9.7 | $2.3.10^{16} \pm 0.5$ | $2.2.10^{16} \pm 0.4$ | 151 | 0.84 |
| **Kiruna** | 8.6 | 11.9 | $2.4.10^{16} \pm 0.4$ | $2.2.10^{16} \pm 0.4$ | 206 | 0.81 |
| **Jungfraujoch** | 13.9 | 9.6 | $1.9.10^{16} \pm 0.3$ | $1.7.10^{16} \pm 0.3$ | 583 | 0.91 |
| **Izaña** | 9.2 | 9.8 | $1.1.10^{16} \pm 0.2$ | $1.1.10^{16} \pm 0.1$ | 256 | 0.74 |
| **Lauder** | 21.7 | 13.0 | $1.7.10^{16} \pm 0.3$ | $1.4.10^{16} \pm 0.3$ | 74 | 0.81 |
| **Arrival Heights** | 1.1 | 16.3 | $1.4.10^{16} \pm 0.4$ | $1.4.10^{16} \pm 0.3$ | 75 | 0.77 |
| **All stations** | 11.5 | 12.1 |  |  | 1345 | 0.93 |




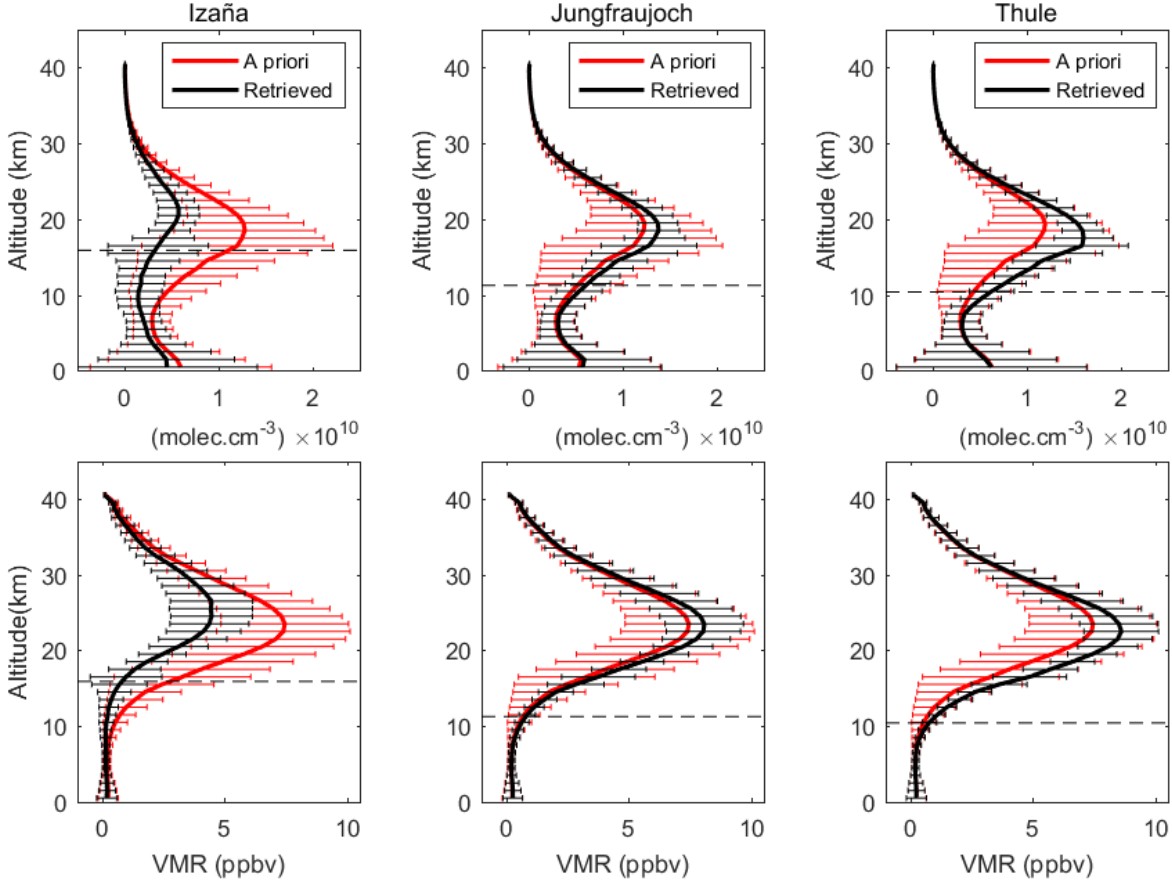

**Figure 1. Example of IASI HNO₃ vertical profiles, for July, at three locations in the northern hemisphere: Izaña (28.3°N, 16.5°W), Jungfraujoch (46.6°N, 8.2°E) and Thule (76.5°N, 69.0°W). The a priori profile and a priori variability (horizontal bars) are represented in red, and the retrieved profile and its error are in black. The concentrations are expressed in molecular density, i.e. molec.cm⁻³ (top panels), and in volume mixing ratio (ppbv, bottom panels). The black dashed line is the altitude of the tropopause, calculated as the lapse-rate tropopause.**

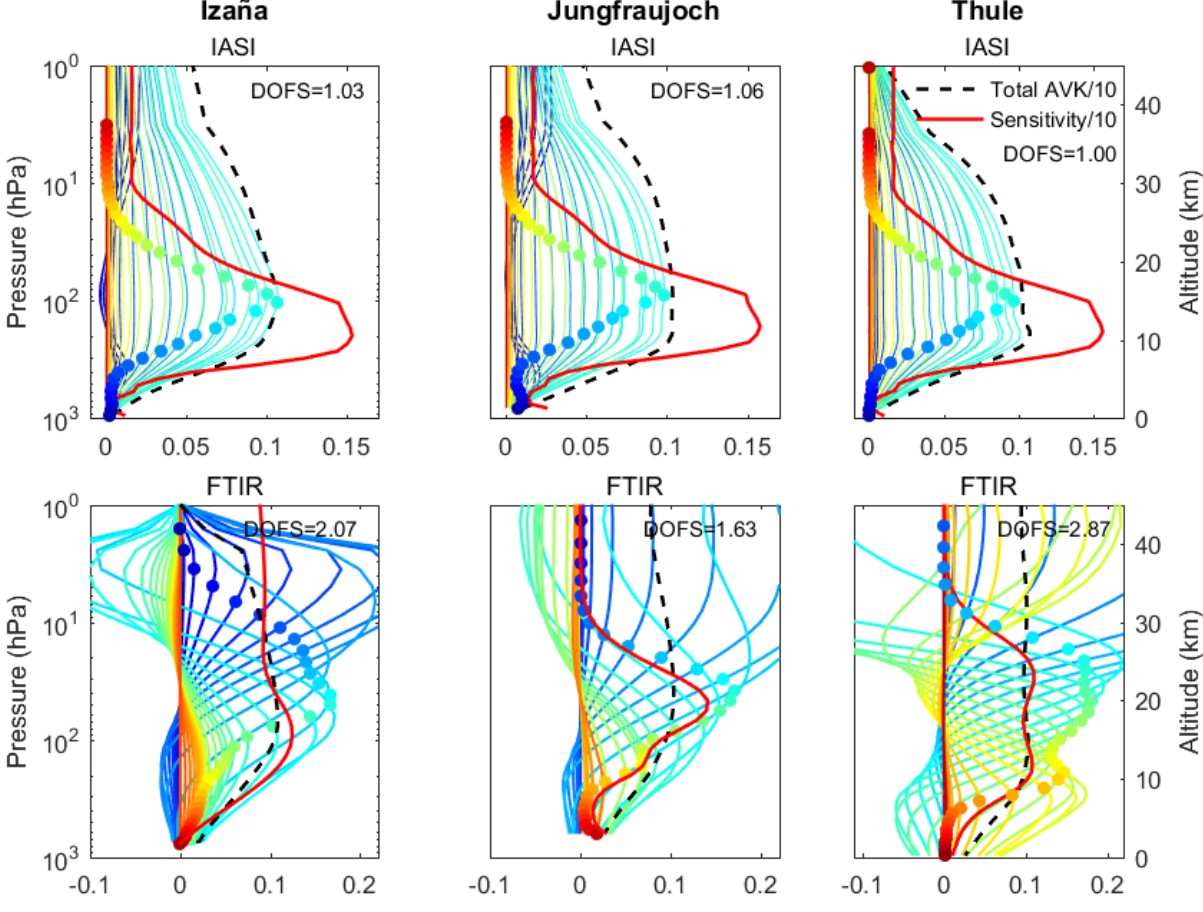

**Figure 2. For three typical latitudes in July: Izaña (tropical, left), Jungfraujoch (mid-latitude, middle) and Thule (polar, right): (top) IASI and (bottom) FTIR averaging kernels for HNO₃ expressed in column units (molec.cm⁻²), The DOFS values are specified on each graph, and the total column averaging kernel is represented by the black dashed line. The red line is the sensitivity (see text for details) and the colored dots represent the altitude of each kernel**





**Figure 3.** For January (left) and July (right) 2011; (top) Global distribution of the DOFS for IASI HNO₃ total columns. The red dots on the left are the location of the stations used for the validation; from north to south: Thule, Kiruna, Jungfraujoch, Izaña, Lauder, Arrival Heights. (middle) Global distribution of surface temperatures (in K). (bottom) Global distribution of the altitude of maximum sensitivity (km) of IASI instrument, in the sense of the total averaging kernel.





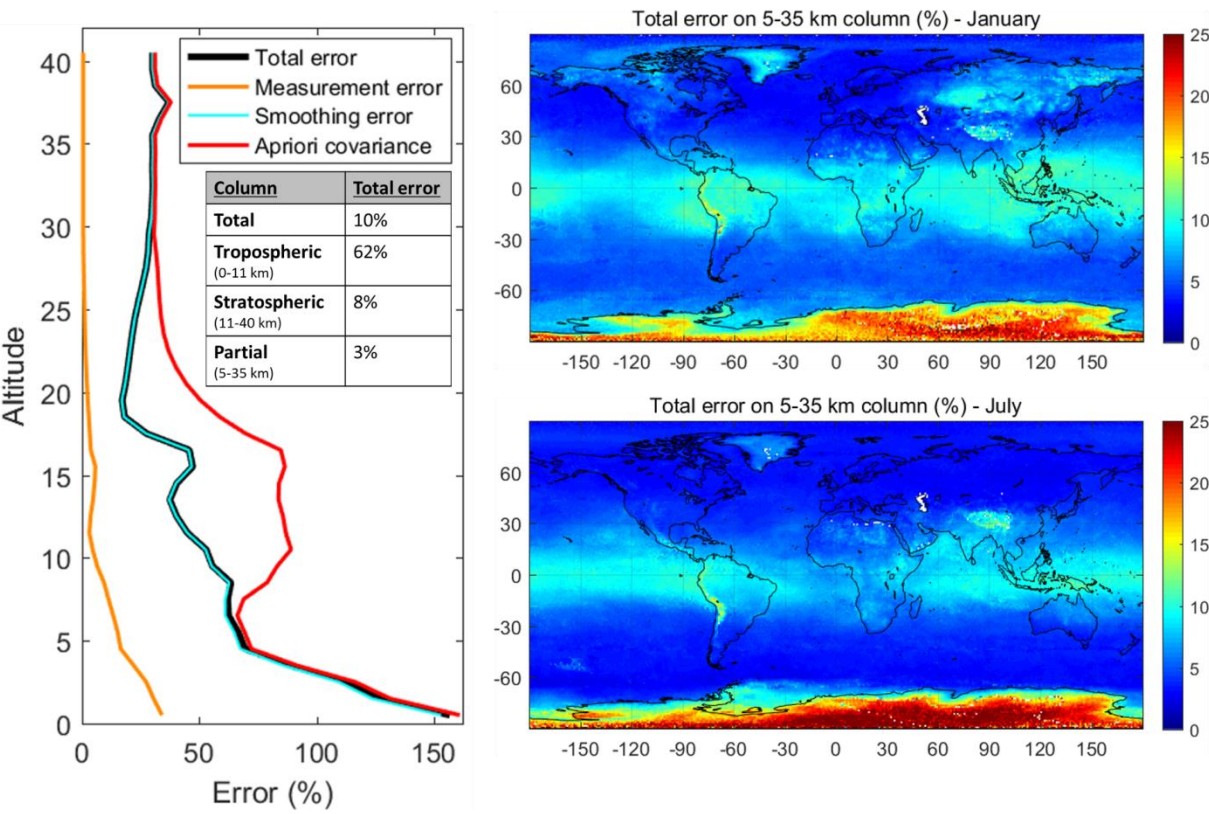

**Figure 4. (left)** Error profiles for FORLI-HNO₃ showed in %. The total retrieval error is in black, the measurement error in orange and the smoothing error in light blue. The a priori variance (square root of the diagonal elements of $S_a$) is in red. Also shown in the table is the total error on different partial columns (tropopause height taken as the lapse-rate tropopause for tropospheric and stratospheric columns). **(right)** Spatial distribution of total retrieval error (%) on the 5-35 km column for January (top) and July (bottom).





**Figure 5.** (left) IASI (red), raw FTIR (black) and regridded and smoothed FTIR mean profiles (green). Also shown are the IASI and FTIR a priori profiles (red and black dashed lines, respectively). (right) Relative differences between the IASI and the FTIR smoothed profiles for each station.





**Figure 6. (top panels)** Comparison of the IASI and the corresponding FTIR partial (5-35 km) columns: the red dots are IASI derived partial columns, the black dots refer to the raw FTIR data and the green dots to the smoothed FTIR. The vertical error bars represent the retrieval total error for each instrument. Also represented (in light grey) is the IASI data set averaged over a small region around the FTIR location (see text for details) and the standard deviation for each data point (grey shaded areas). The IASI a priori partial column is represented by the dashed grey line. **(bottom panels)** Relative differences (dots), bias (dashed line) and standard deviation (dotted lines) between the two data sets for each station: in black, the difference between IASI and the raw FTIR data, in green, the difference between IASI and the smoothed FTIR data.





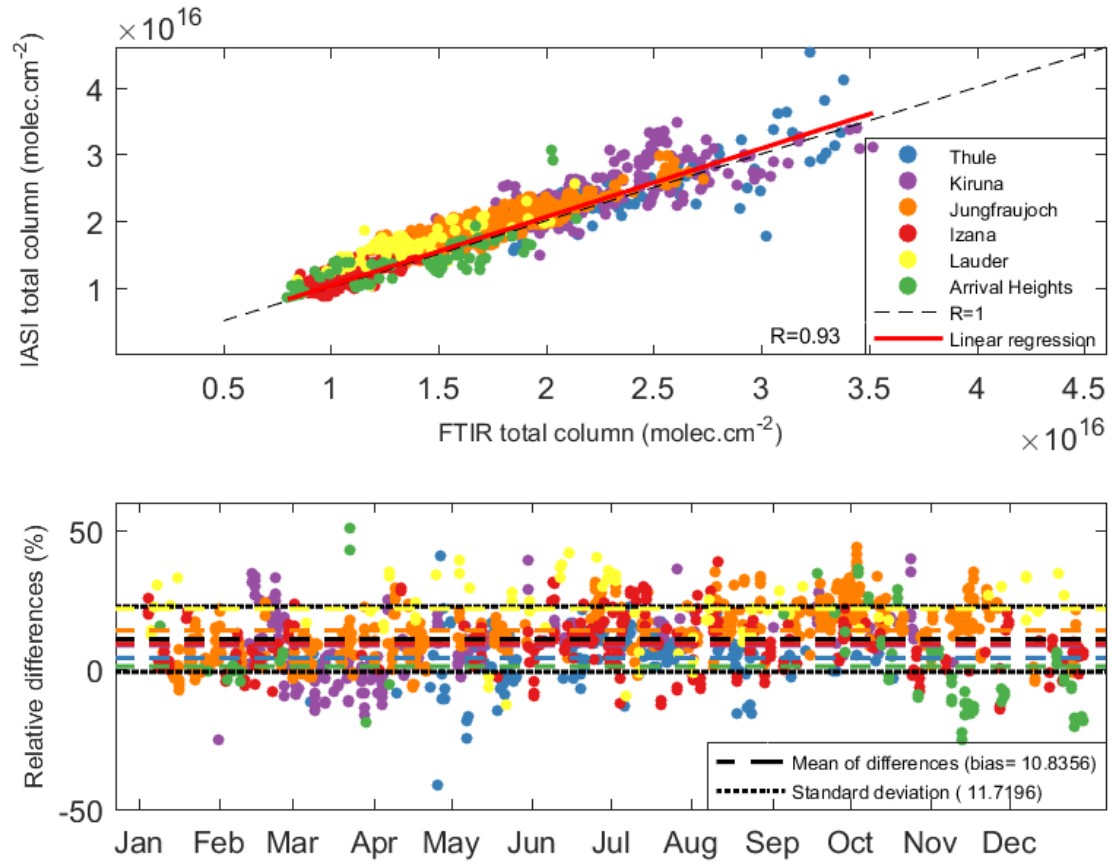

**Figure 7. (top) Comparison of the smoothed FTIR and IASI partial (5-35 km) columns for all stations considered, for the year 2011. Also shown is the correlation coefficient value. (bottom) Time series of the relative differences between IASI and FTIR total columns (calculated as ((IASI-FTIR)/FTIR)*100) (%). Also shown are the values of bias for each station (colored dashed lines) and the bias and standard deviation when considering all stations together (black dashed and dotted lines, respectively).**





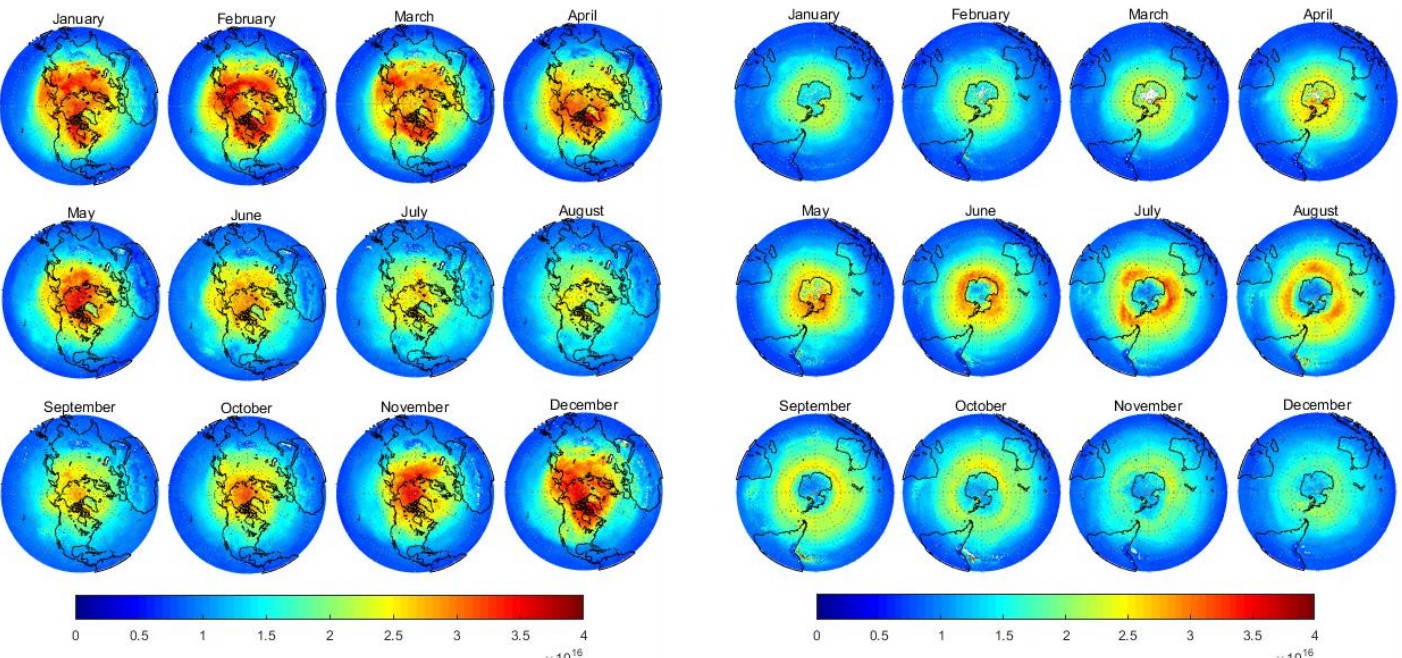

**Figure 8. Monthly global distributions of IASI HNO$_3$ total columns in 2011 in the Northern Hemisphere (left) and the Southern Hemisphere (right). Columns are expressed in molec.cm$^{-2}$.**

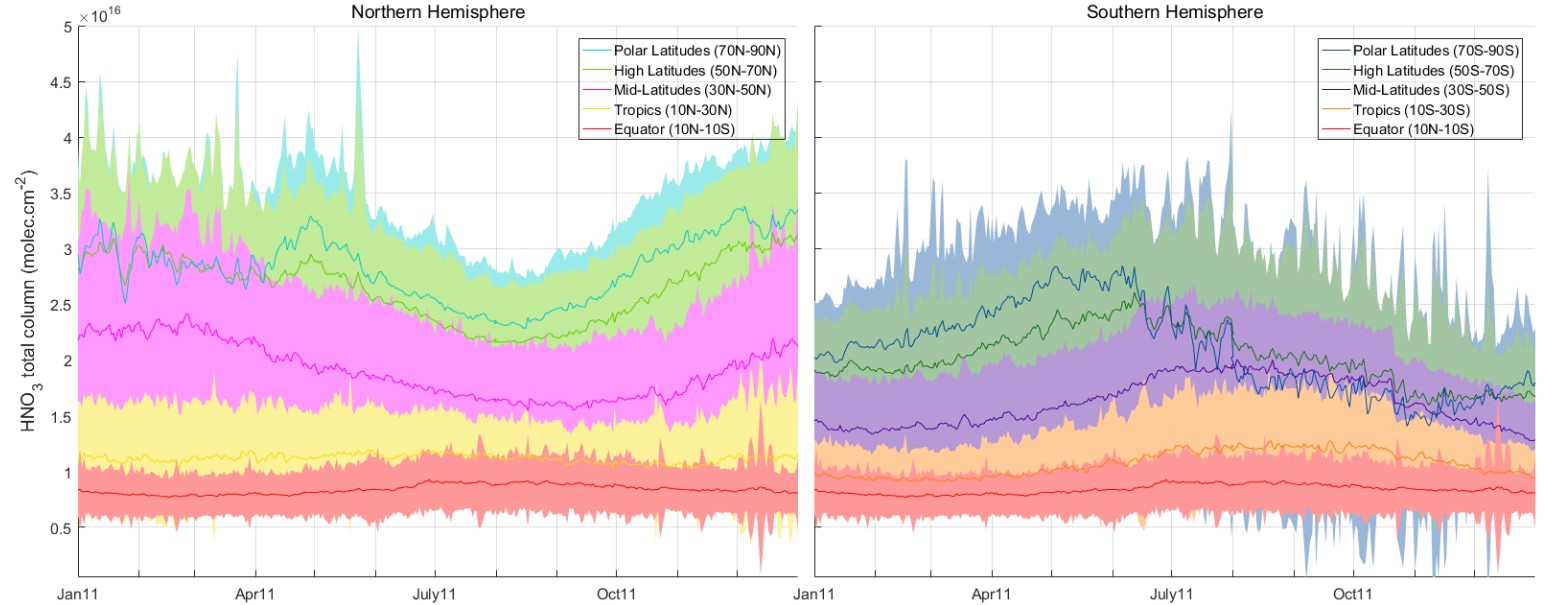

**Figure 9. Seasonal variability (daily mean) and daily variability (shading, calculated as the standard deviation of the daily mean) of the total column for 9 latitudinal bands of 20°, each shown with a different color.**