# Peer review of "First characterization and validation of FORLI-HNO3 vertical profiles retrieved from IASI/Metop"

_Atmospheric Measurement Techniques, 2016_

## Referee Comment (RC1) · E. Remsberg (Referee) · 21 Jul 2016

Review from Ellis Remsberg of AMTD manuscript 2016-207 by Ronsmans et al.:

**General comments:**

The present manuscript extends a similar, earlier analysis of column measurements of HNO3 from IASI as reported by Wespes et al. (ACP, 2009). It differs from that earlier work by showing quantitative comparisons with data from six representative NDACC stations that span the latitude range of the IASI measurements. Those comparisons with IASI and the seasonal HNO3 distributions are also for a different year-2011. I find that the analysis approach, figures, and discussion of findings to be of very good quality, and I recommend publication after some very minor revisions.

**Specific comments:**

Introduction, p. 3, line 68—I would argue that it is difficult to use such column data for quantitative studies of the HNO3-ozone cycles.

Section 5, p.10, line 303—The overestimation by IASI of 47% is rather large in the lower stratosphere for Lauder. I note that the *a priori* profile for HNO3 comes from your chemistry transport model at up to 15.6 km altitude. Might this be a cause of the rather large bias?

Section 5, p. 11, line 308—Please add a few sentences about the hypotheses of Dufour et al. (2012). For example, I note that they discuss a likely interdependence for the retrieved partial ozone columns between the stratosphere and the UTLS regions.

Section 6, p. 13, lines 385-387—Day/night differences in HNO3, due to photochemistry, ought to be small in the tropics below about 25 km. Is this the issue that you are referring to?

Section 7, p. 14, lines 433-435—This finding is important and may be one cause of the positive bias of 47% that you found at Lauder (see Section 5 comment above).

Technical comments:

Abstract, p. 1, line 2—I recommend "comprehend" instead of apprehend.

Introduction, p. 2, line 52—AURA operations began in 2004.

Section 3, p. 6, line 180—Say instead "This plot shows..."

---

## Referee Comment (RC2) · Anonymous Referee #2 · 3 Aug 2016

This study compares IASI HNO3 data for 2011 with correlative data from multiple ground-based NDACC stations from across the globe. The results are well founded and are relevant to an AMT reader, and the paper is very well written. I would recommend the paper for publication pending some minor corrections, detailed below.

**Minor issues**

Line 67 – Does Wang et al. 2007 actually state that ACE-FTS is measured with a 3% accuracy, or does ACE-FTS HNO3 agree with correlative data from other instruments to within 3%? Because ACE-FTS does not have official accuracy/precision estimates for their data sets.

Lines 69-70 - Please explain what is meant by "less used".

Lines 81-82 – "suffers" might be a bit harsh. I would suggest rewording to something like, at the time of Wepse et al. 2009, the FLORI algorithm only allowed for total column retrievals, and it wasn't possible to do a rigorous validation study.

Line 89 – Why are you limiting to just one year of data? Please explain why 2011 was chosen. As well, perhaps in the Conclusions section (or wherever you feel it fits best), it would useful to mention if 2011 are representative of other years, or how results from other years might differ.

Lines 134-135 – Hurtmans et al. 2012 does not actually go over how sensitivity is calculated, nor the total column averaging kernel. There should be a quick one or two sentences (either here or in the discussion of Fig. 2) on how these are calculated, just so there's no confusion.

Line 138 – Please briefly explain how water vapour is accounted for in the retrieval, and what is the effect on HNO3 uncertainty?

Line 151 – Please explain what is meant by "suspect" averaging kernels.

Figure 1 caption – I'd suggest changing "variability" to " $1\sigma$  variation" (i.e. square root of the diagonal of the covariance), unless this is not what is shown.

Line 179 – "close to one" is vague, please give a typical range.

Lines 180-187 –How do increased surface temperatures increase DOFS when there is no or little sensitivity at near-surface altitudes (it is not immediately clear to me)? Does this mean that where you have higher surface temperatures you also have increased sensitivity to HNO3 at near-surface altitudes? If this is the case, can you please show this in a plot?

Lines 242-243 – In what way are the data sets updated?

Lines 283-287 – as discussed above, this discussion may be more useful when Fig 2

is first mentioned

Line 364 – here and table 4 state that the overall mean difference is 11.5%, but the legend in Figure 7 states that it is 10.8%. Which value is correct?

Lines 373-374 – It should be stated that the biases at both stations are still within the uncertainties of both IASI and the FTIRs. Hence, the "different behaviour" isn't of great concern.

Line 382-382 – How is one year of data enough to be able to reliably comment on the trend in the bias?

Line 488 - What factors in the retrieval codes lead to biases?

Technical issues

Line 10 - DOFS should have a space after it, and should be defined.

Line 68 - "the ODIN instrument" should be "the SMR instrument on the Odin satellite".

Figure 3 - colour bar labels should be on the right hand side, next to the colour bars. The labels on the left hand side should be "Latitude (deg)", and the x-axis should be labelled as longitude.

Figure 4 - legend label "covariance" should read "variance".

Line 255 – by "regardless" do you mean "independent of"?

Line 299 – should be "use the method of Rodgers..."

Figure 7 – the coloured dashed lines are not easily visible (and perhaps could be omitted).

Figure 9 – Many of the shaded regions are covered, diminishing their usefulness. I would suggest perhaps not shading, instead using coloured dashed lines at the extremes.

---

## Author Comment (AC1) · 9 Sep 2016

Response to Ellis Remsberg

We would like to thank E. Remsberg for his constructive and positive comments. The comments and proposed corrections have been taken into account and helped improving the paper. Each comment has been addressed as detailed hereafter.

**Specific comments**

   *(1) Introduction, p. 3, line 68—I would argue that it is difficult to use such column data for quantitative studies of the HNO3-ozone cycles.*

We understand that you are referring here to the different vertical sensitivity of IASI for ozone ($O_3$) and nitric acid ($HNO_3$). This is indeed a good point. We are confident, however, that this would not be such an issue, considering that there is quite a good vertical sensitivity for $O_3$ (DOFS ~ 3-4) allowing the distinction of a stratospheric column (sometimes even two independent columns within the stratosphere), and that most of the information about $HNO_3$ (whether considering a total or a stratospheric column) is located in the same lower part of the stratosphere. This type of parallel study for $O_3$ and $HNO_3$ was already conducted by Wespes et al. (2012).

However, the difference in vertical sensitivity would of course need to be adressed when analysing the two species in parallel.

   *(2) Section 5, p.10, line 303—The overestimation by IASI of 47% is rather large in the lower stratosphere for Lauder. I note that the a priori profile for HNO3 comes from your chemistry transport model at up to 15.6 km altitude. Might this be a cause of the rather large bias?*

First of all, it should be noted that Lauder and Arrival Heights data sets have been corrected after the manuscript was published; this was due to errors in the averaging kernels for both these stations. The comparison has been redone and all figures and tables have been updated with the correct values. No major differences were found but still the correction of the AvK improved slightly the comparison. The maximum relative difference in Lauder is now reduced to 37.2%., as can be seen below with the updated version of Table 3 for the profile validation (page 24, Table 3):

| | Minimum (%) [altitude (km)] | Maximum (%) [altitude (km)] |
|---|---|---|
| Thule | 0.4 [22] | 12.5 [13] |
| Kiruna | -0.1 [24] | 18.0 [13] |
| Jungfraujoch | 0.1 [37] | 25.8[12] |
| Izaña | 0.21 [2] | 45.0 [13] |
| Lauder |  0.7 [39] |  37.2 [12] |
| Arrival Heights |  0.3 [4] |  1.8 [13] |

Despite of this and as you point out, the differences in Lauder and Izaña are still quite large. The a priori profiles do indeed seem to have quite a large impact on the IASI retrieval (see George et al. 2015 for the example of CO), hence on the comparison between the two data sets. It should also be noted, as is mentioned in the paper, that IASI uses a common a priori profile for all measurements, whereas FTIR stations use an a priori profile adapted to the

region. In our work, the influence of the a priori was examined with the example of Izaña and a short sentence was added in the conclusions to account for the fact that it might also explain the large differences observed in Lauder: "However, as was shown by the comparison at Izaña, the influence of the a priori profile on the validation can be quite large, and the application of a common a priori profile to both measurements largely improves the comparison. The difference in the a priori profiles could also explain in part the differences found at other stations (Lauder, for example)." (page 15, line 443-446).

> *(3) Section 5, p. 11, line 308—Please add a few sentences about the hypotheses of Dufour et al. (2012). For example, I note that they discuss a likely interdependence for the retrieved partial ozone columns between the stratosphere and the UTLS regions.*

A few sentences about the hypotheses of Dufour et al. (2012) have been added to complete the paragraph on this open question: "While some hypotheses have been brought forward by Dufour et al. (2012), the exact reason for that particular feature of FORLI for both $HNO_3$ and $O_3$ retrievals is not clear. The loose constraint applied for the retrieval at these altitudes, combined with a lack of vertical sensitivity, could be one reason to explain the overestimation in the UTLS, as it might be that the UTLS concentrations are overestimated to compensate for lower values in the rest of the profile (Dufour et al., 2012). A more in-depth analysis would, however, be needed to assess this matter in more details." (page 11, lines 311-316).

> *(4) Section 6, p. 13, lines 385-387—Day/night differences in HNO3, due to photochemistry, ought to be small in the tropics below about 25 km. Is this the issue that you are referring to?*

What we are referring to here is the fact that the diurnal variability (i.e. the $3\sigma$ standard deviation - grey shaded areas) is of the same magnitude as the retrieval error. Strictly looking at the figure, it is thus hard to assert that there is any diurnal variability. It means that the IASI measurements would not allow the monitoring of diurnal variability, even if there were any, considering that the error is too large.

> *(5) Section 7, p. 14, lines 433-435—This finding is important and may be one cause of the positive bias of 47% that you found at Lauder (see Section 5 comment above).*

See Comment 2.

George, M., Clerbaux, C., Bouarar, I., Coheur, P. F., Deeter, M. N., Edwards, D. P., … Worden, H. M. (2015). An examination of the long-term CO records from MOPITT and IASI: Comparison of retrieval methodology. *Atmospheric Measurement Techniques*, *8*(10), 4313–4328. http://doi.org/10.5194/amt-8-4313-2015

Wespes, C., Emmons, L., Edwards, D. P., Hannigan, J., Hurtmans, D., Saunois, M., … Wennberg, P. O. (2012). Analysis of ozone and nitric acid in spring and summer Arctic pollution using aircraft, ground-based, satellite observations and MOZART-4 model: Source attribution and partitioning. *Atmospheric Chemistry and Physics*, *12*(1), 237–259. http://doi.org/10.5194/acp-12-237-2012